# Modification of Quaternary Clays Using Recycled Fines from Construction and Demolition Waste

**Roumiana Zaharieva** [1,*] **, Daniel Evlogiev** [2] **, Nikolay Kerenchev** [3] **and Tsveta Stanimirova** [4]

1 Department of Building Materials and Insulations, Faculty of Structural Engineering, University of Architecture, Civil Engineering and Geodesy, 1 Hristo Smirnenski Blvd., 1046 Sofia, Bulgaria

2 Centre of Competencies "Clean&Circle", Faculty of Structural Engineering, University of Architecture, Civil Engineering and Geodesy, 1 Hristo Smirnenski Blvd., 1046 Sofia, Bulgaria; d.evlogiev_fce@uacg.bg

3 Department of Geotechnics, Faculty of Transportation Engineering, University of Architecture, Civil Engineering and Geodesy, 1 Hristo Smirnenski Blvd., 1046 Sofia, Bulgaria; kerenchev_fte@uacg.bg

4 Department of Mineralogy, Petrology and Economic Geology, Faculty of Geology and Geography, Sofia University St. Kliment Ohridski, 15 Tzar Osvoboditel Blvd., 1000 Sofia, Bulgaria; stanimirova@gea.uni-sofia.bg

* Correspondence: zaharieva_fce@uacg.bg; Tel.: +359-8-8463-6212

**Abstract:** Foundation of buildings in soft soil such as quaternary clay is often associated with difficult compaction, settlement, non-uniform and/or excessive deformation, and unsatisfactory shear resistance. The present study aims to assess the possibility of using recycled fines from construction and demolition waste, such as mechanically treated gypsum and waste concrete powder (WCP), instead of ordinary binders or industrial waste, in the stabilization of quaternary clay. A detailed characterization of soil components is presented. Seven mixes with various proportions of gypsum and WCP are prepared. Main geotechnical parameters of the modified soil are studied by applying standardized methods with a few deviations. XRD analysis and pH measurements are performed. It was found that the effect of 5% to 20% recycled di-hydrate gypsum is limited to improvement in moist soil compatibility. A gypsum content of 10% positively impacts soil cohesion and the oedometer modulus. WCP is an active component, containing non-hydrated cement, portlandite, calcite and calcium silicates hydrate. As a result, by adding 5% of WCP only, significant improvement can be achieved: greater soil cohesion, reduced deformability and higher UCS. When 5% of recycled gypsum is also added, soil cohesion is further improved because of ettringite formation.

**Keywords:** clay; CDW; recycled gypsum; recycled concrete powder; soil compaction; shear resistance; soil deformation; the oedometer modulus; UCS

## 1. Introduction

From a geotechnical point of view, the foundation of building and transport facilities in soft clay is often associated with challenges such as difficult compaction, settlement, non-uniform and or excessive deformation, as well as unsatisfactory shear resistance. Soil replacement is sometimes applied, but it is far from being a sustainable solution because of both high costs and a high environmental footprint. Saturated clays are very often the object of replacement, but saturated quaternary clays are sometimes also problematic and classified as deformable or susceptible to deformation. Quaternary clays are widely presented in Bulgaria—in the Sofia valley, their deposits are at a depth of 4 to 12 m [1].

In situ soil stabilization is the best option to solve the workability (compatibility), bearing capacity and deformability issues of clayish soils. The Bulgarian Road administration specifies various methods for soil stabilization: (a) chemical, by using polymeric substances or different binders: cement, lime and hydraulic road binders and their combinations; (b) mechanical stabilization by using rock materials, milled asphalt or slag; (c) a combination of those methods [2].

The utilization of industrial waste and by-products (such as bottom ashes, fly ashes, slag and phosphogypsum) is a very attractive opportunity for soil stabilization [3,4], but their applicability is relatively limited, because of logistic and market challenges—for example, in Bulgaria, there is only one location of accumulated blast furnace slag (in the vicinities of Sofia), only one source of phosphogypsum (400 km away from Sofia) and the main sources of bottom ashes and fly ashes are 250 km away from Sofia.

In contrast, construction and demolition waste (CDW) are abundant sources of materials all over the country, proven to be suitable for different construction purposes. Concrete CDW is among the largest waste streams—its share in demolition waste in Europe is estimated to be 50–55% [5], thus it amounts to more than 450 million tones [5,6] per year. In Bulgaria, concrete CDW represents approximately 25% of total amount of CDW in Bulgaria [7]. So, approximately 438,264 tons of concrete CDW was generated in 2020 [8]. The implementation of CDW for soil stabilization would contribute to both waste recovery and natural resource preservation [3]. For instance, recycled concrete crushed stone can be used for mechanical stabilization of clayish soil, while recycled waste concrete powder (WCP), because of its hydraulic activity [9,10], has potential for chemical stabilization—it was established that when clay containing $SiO_2$ and $Al_2O_3$ is in an alkaline environment, there is a possibility to participate in hydration and pozzolanic reactions, which leads to the formation of calcium hydroxide (CH) and calcium silicates hydrate (C-S-H) as well as calcium aluminate hydrate (C-A-H) and calcium aluminate silicate hydrate (C-A-S-H) gel [11], thus increasing the cohesion, friction angle and water impermeability of the soil. The mineral composition of the soil is very important—the most active components are fine-grained quartz and the feldspars. The chemical stabilization effect is more pronounced when the soil pH is high, its organic content is limited and the water-soluble salts are in small amounts [12].

WCP might have a different composition in the function of primary concrete mix design and environmental impacts, but in general it consists of hydrated cement, non-hydrated cement, quartz, carbonates, dolomites and feldspars originating from the aggregates in primary concrete [9,10]. Depending on their proportions, WCP might be quite active and then the chemical processes will be more intensive, or can be rather inert and then the physical effects of its use in soil stabilization will be predominant. The active WCP can be used for replacement of Portland cement and hydraulic road binders in soil stabilization. A serious reduction in greenhouse gases (GHG) emission will be achieved, because 1 ton of Portland cement is associated with 750–800 kg of $CO_2$ eq., while the recycling of concrete generates a much smaller amount of GHG [10,13,14]. No data on using WCP for stabilization of quaternary clay were identified.

Gypsum CDW is in a much smaller amount than concrete. However, according to [15], the amount of waste gypsum produced every year worldwide is as high as 80 million tons; for Bulgaria, it is estimated to 14,000 tons, generated from various activities (construction, repair and rehabilitation works and demolition). So far, gypsum has not been utilized and it is mainly stored in dedicated places, because the disposal of gypsum waste requires special attention—gypsum waste is not inert and requires disposal at landfills for non-hazardous waste (as those for municipal waste), but must not be disposed with municipal waste because there is a risk of methane formation. At the same time, gypsum from CDW is usually di-hydrates ($CaSO_4 \cdot 2H_2O$) and can be successfully recycled—mechanically to a powder as a filler or thermally to a semi-hydrate ($CaSO_4 \cdot 0.5H_2O$), i.e., converted again to construction gypsum. There are some studies indicating that when di-hydrates and/or semi-hydrates (natural or recycled) are added to non-gypsiferous soil, an improvement in the mechanical behavior of subgrade soil such as silty clay can be achieved [16,17]. However, gypsum-stabilized soils are not durable in an immersed state and when subjected to alternating freezing and thawing [3,18], so they can be applied in particular cases only. The combination of gypsum and lime or cement binders is usually implemented for stabilization of soft (deformable) clays [3]. Recycled gypsum has also been successfully implemented [16]. The greater the amount of these binders, the better the stabilized soil

performance is. The effect is due to several physicochemical processes: water absorption of gypsum leading to the so-called "drying out" of the clay, flocculation, formation of ettringite, calcium silicate hydrate (C-S-H), calcium aluminate hydrate (C-A-H) and calcite ($CaCO_3$).

The main purpose of this study is to assess the possibility of soil stabilization by recycled fines (powders) from concrete and plasterboard CDW instead of ordinary binders and raw materials. The following necessary steps were identified: (a) physical, chemical and mineralogical characterization of the soil components in order to assess their reactivity potential [1,2,12]; (b) optimization and modifying of the fines content, since the recommended quantity of gypsum varies a lot [3,18]; (c) determination of the main geotechnical parameters of the soil in order to assess the stabilization effect [3]; (d) study of the interaction processes in soil to clarify the observed engineering phenomena. This study aims to provide feasible solutions to improve the mechanical behavior of quaternary clays, and to contribute to the reduction in the environmental footprint of the soil stabilization process and to the increase in concrete and gypsum CDW recovery.

## 2. Materials and Methods

### 2.1. Quaternary Clays

Soil from the region of the Sofia valley with mass of 300 kg was brought to the geotechnical laboratory of UACEG. The material was classified as quaternary clay and is in a disturbed condition. Its mineral composition according to the semi-quantitative XRD analysis as per 2.5.1. consists mainly of quartz (ca. 44%), feldspars (ca. 41%), amphiboles (ca. 7%) and muscovite and clay minerals (ca. 8%)—Figure 1. The results correspond to the findings in [1].

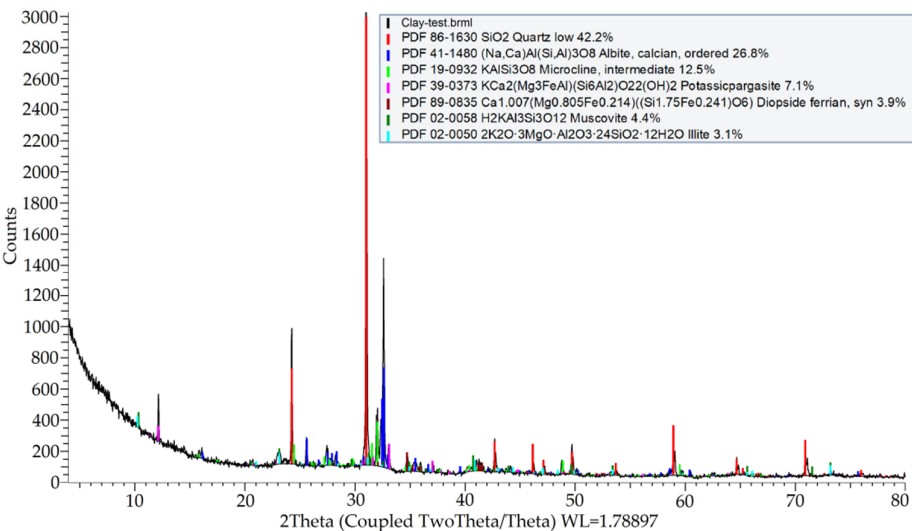

**Figure 1.** XRD pattern of the used soil—quaternary clay from the region of Sofia.

The tested soil can be characterized as silty clay (SiCl) according to ISO 17829-4. Due to the test procedure requirements, the soil was sieved and the fraction 0/2 mm was used for further investigation. The main soil parameters were determined as per relevant standard methods (Table 1).

**Table 1.** Quaternary clay properties.

| No | Parameter | Unit | Method | Value |
|---|---|---|---|---|
| 1 | Bulk density, $\rho$ | $(Mg/m^3)$ | BDS EN ISO 17892-2:2015 [19] | 2.03 |
| 2 | Dry density, $\rho_d$ | $(Mg/m^3)$ | BDS EN ISO 17892-2:2015 [19] | 1.71 |
| 3 | Specific density, $\rho_S$ | $(Mg/m^3)$ | BDS EN ISO 17892-3:2016 [20] | 2.65 |
| 4 | Porosity, $n$ | (%) | Calculation based on parameters determined as per BDS EN ISO 17892-1, 2, 3 [19–21] | 35.6 |
| 5 | Plastic limit, $W_p$ | (%) | BDS EN ISO 17892-12:2019 [22] | 23.1 |
| 6 | Liquid limit, $W_L$ | (%) | BDS EN ISO 17892-12:2019 [22] | 42.5 |
| 7 | Plasticity index, $I_p$ | (%) | BDS EN ISO 17892-12:2019 [22] | 19.5 |
| 8 | Passing sieve 0.063 mm | % | BDS EN ISO 17892-4:2016 [23] | 58.5 |

The particle size distribution of quaternary clays was determined by EN ISO 17892-4:2016 [23] and is illustrated in Figure 2.

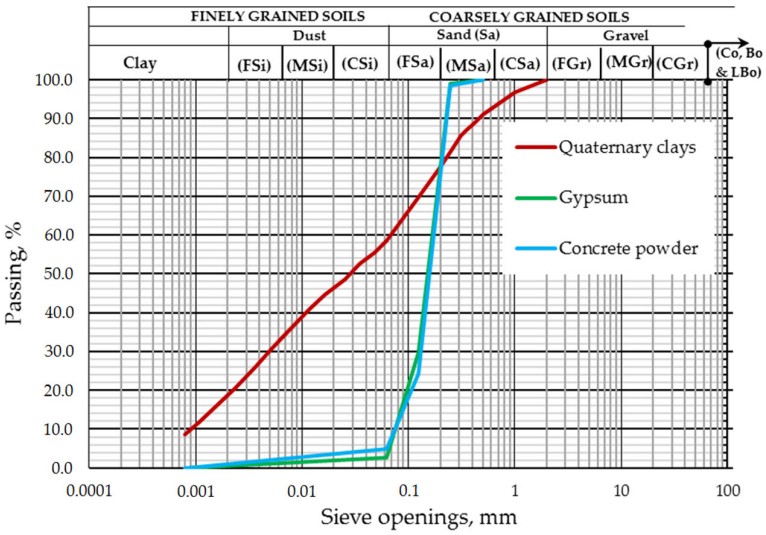

**Figure 2.** Grading curves of the used quaternary clay fraction, recycled gypsum and WCP.

According to the Technical Specification of the Bulgarian Road Administration [2], the tested soil belongs to the soils of group A-7-6, and is not appropriate for road embankments. After being stabilized, this type of soil can be used for road embankment works.

*2.2. Recycled Gypsum*

Gypsum used for clay stabilization was obtained by mechanical recycling only of plasterboards type "A", manufactured by KNAUF Bulgaria in compliance with BDS EN 520:2004 + A1:2009 and BDS EN 520:2004 + A1:2009/NA:2014. During recycling in laboratory conditions, the cardboard was removed manually, and the plaster core was crushed and then milled in Los Angeles apparatus as per BDS EN 1097-2:2020, applying 1000 rotations [24]. The fraction 0/0.5 mm was used for clay stabilization. Gypsum powder was dried out for 1 h in a ventilated oven at 105 °C. According to XRD analysis, as per Section 2.5.1., the recycled gypsum consists mainly of di-hydrates (ca. 93%) and a small amount of bassanite (nearly 4%). A small amount of calcite (ca. 3%) has also been identified—Figure 3. The recycled gypsum grading analysis was performed according to BDS EN 933-1:2012 [25]. The grading curve is presented in Figure 2. The recycled gypsum has a specific density, $\rho_S$, of 2.26 $Mg/m^3$, determined according to BDS EN 1097-6:2022 [26].

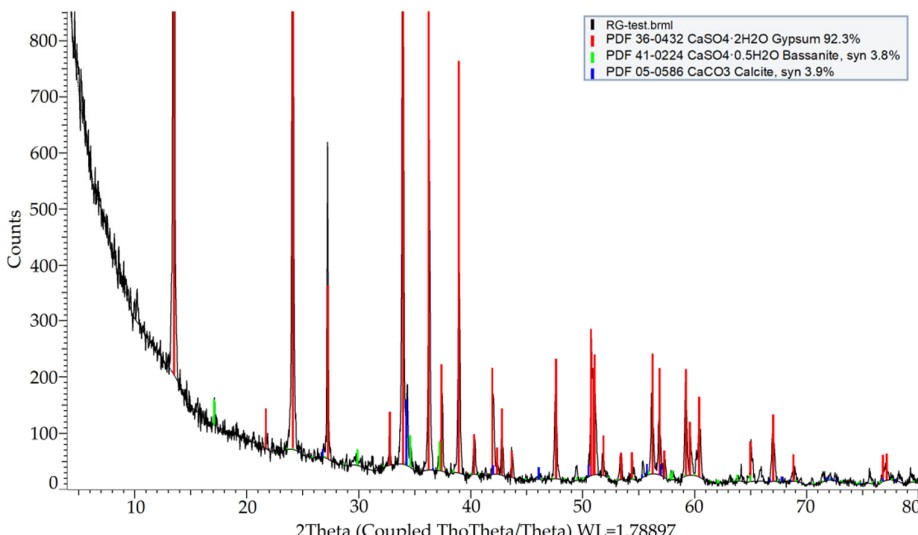

**Figure 3.** XRD pattern of gypsum recycled from plasterboard CDW.

### 2.3. Concrete Powder

The primary concrete of concrete CDW originates from high-strength railway sleepers. A recycled concrete fraction 22.4/63 mm was delivered to the laboratory by the recycling company. First crushing and milling were executed in Los Angeles apparatus at 2000 rotations. Final milling was performed in micro-Deval apparatus as per BDS EN 1097-1:2011 at 20,000 rotations [27]. The fraction 0/0.5 mm was used for clay stabilization. The mineral composition of the used WCP determined by XRD analysis described in Section 2.5.1. consists of 28% of quartz, 33% feldspars, 10% mica, 15% dolomite, calcium silicate hydrates such as tobermorite (10%), calcite (ca. 2.6%) and portlandite (1.6%)—Figure 4. The WCP grading analysis was performed according to BDS EN 933-1:2012 [25]. The grading curve was presented in Figure 2. The recycled WCP has a specific density, $\rho_S$, of 2.68 Mg/m$^3$, determined according to BDS EN 1097-6:2022 [26].

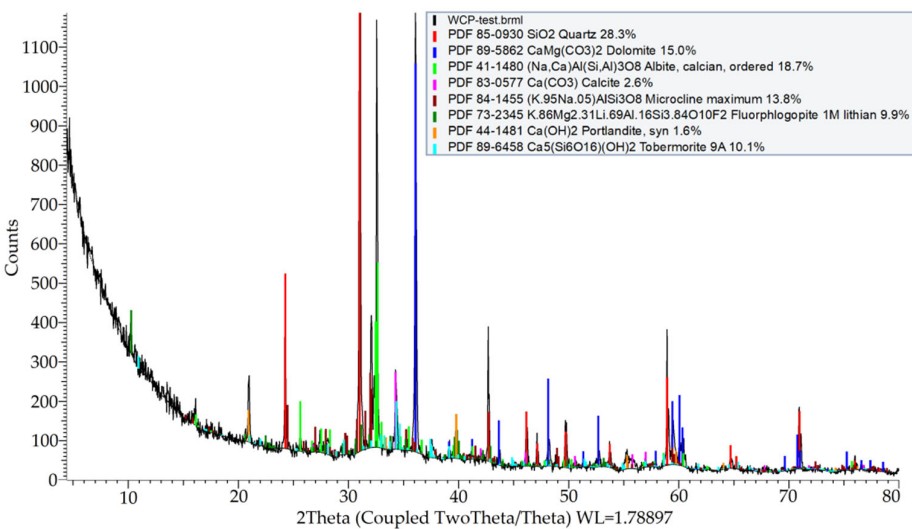

**Figure 4.** XRD pattern of WCP from recycled concrete sleepers.

### 2.4. Mix Design

Different modifications of the clay were investigated. The mix design proportions are based on different studies and correspond to 0 to 20 percent for the recycled materials. The mix design proportions are presented in Table 2.

**Table 2.** Composition of modified soil.

| No | Mix Designation | Quaternary Clays, %wt | Recycled Gypsum, %wt | WCP, %wt |
|---|---|---|---|---|
| 1 | Cl100 | 100 | 0 | 0 |
| 2 | Cl95-G5 | 95 | 5 | 0 |
| 3 | Cl90-G10 | 90 | 10 | 0 |
| 4 | Cl80-G20 | 80 | 20 | 0 |
| 5 | Cl85-G10-WCP5 | 85 | 10 | 5 |
| 6 | Cl90-G5-WCP5 | 90 | 5 | 5 |
| 7 | Cl95-WCP5 | 95 | 0 | 5 |

*2.5. Sample Preparation and Test Methods*

2.5.1. XRD Analysis

X-ray diffraction phase analysis (XRD) was performed on the three soil ingredients (quaternary clay, recycled gypsum and WCP) and their mixes. Samples of approximately 50 g of modified soil were taken from each core of cylinders used for UCS determination (2.5.6.) at the age of 3 days. The samples were kept sealed and were prepared for XRD study at the age of 2 months. XRD was performed using Powder X-ray Diffractometer Bruker D8 Advance with a LynxEye detector and with Co Kα radiation, vertical θ/θ goniometer, and a step size of 0.02 (2θ). Diffracplus EVA software using ICDD-PDF2 crystallographic database was used for the structural data and semi-quantitative phase analysis [28]

2.5.2. Measurement of Potential of Hydrogen (pH)

The pH of the soil components (clay, recycled gypsum, WCP and water) were important for the chemical interactions between them after mixing [29]. The pH of the aqueous extract from the materials was measured to determine their acidity and alkalinity. The extract was prepared with deionized water at a solid–liquid ratio of 1:10. The analysis was performed 2 h after the preparation of the extract, using a calibrated pH meter, model "Hanna 211" operating in the pH range 0–14 and with an accuracy of 0.01. Three tests for each component and mix were performed. The average value and standard deviation were calculated.

2.5.3. Determination of Optimum Water Content

The optimum moisture content (OMC) for achieving the maximum dry density was determined according to standard Proctor test method [30]. The sample preparation was based on the following procedure: the clay had been wetted for 12 h prior to testing, while the modifying compounds (gypsum, WCP or their combination) were added just before performing the Proctor compaction, in order to avoid premature chemical reactions (if any) and destruction of the new products resulting from those reactions. Based on the described procedure, five remolded samples with different water contents for every mixture were prepared. The compacted cylindrical samples are 10 cm in diameter and 12 cm in height.

2.5.4. Direct Shear Test

As the shear failure is predominant in soils, direct shear tests (DST) of the samples were conducted. Based on the DSTs, the shear resistance and the shear parameters of the soil and soil mixtures were obtained. DSTs were conducted according to [31] on samples with a constant water content of 18%. An exception to the ordinary test method was that the sample was not immersed in water, taking into consideration the presence of gypsum which might be affected by the water. Actually, gypsum-modified quaternary clays are not to be used in immersion conditions, so the test reproduced the real conditions, since at the water content which is close to the OMC, clayish soils possess both crystalline and viscous (colloidal) bounds. For the quantitative assessment of these inner bounds, the friction angle (φ in°) and the cohesion (c, in kPa) were obtained.

2.5.5. The Oedometer Test

The oedometer modulus gives information on the deformability of the soil in limited or zero lateral deformation. The oedometer test is also informative on soil particle rearrangement, void ratio and indirect bearing capacity correlation. The oedometer test was conducted according to [32] on samples prepared with 12% water content, not saturated at the beginning of the test as this test is usually performed. The main reason for this test modification was the presence of gypsum particles, which might be dissolved, causing structural defects which would not exist in the in situ stabilized soil.

2.5.6. Unconfined Compressive Strength Test

The unconfined compressive strength (UCS) was applied as an indirect criterion for the bearing capacity of modified clay, although the compression failure was not a typical failure mode in soil. UCS is a quantitative parameter representative of the structural modifications of soil (grading, porosity), as well as for the newly formed products from chemical reactions between quaternary clay, recycled gypsum and WCP. UCS was determined as per [33], using three samples, compacted by the standard Proctor procedure [30] at OMC, wrapped immediately after preparation to avoid water exchanges. The testing was performed at the age of 3 days. The average value of the three is reported for each mix.

**3. Results and Discussion**

*3.1. Reactivity Potential*

The pH of each ingredient and of the modified clay mixes is given in Table 3. WCP is characterized by a high alkaline reaction (pH = 11.78), which indicates its reactivity potential. It was found that the addition of gypsum only marginally modifies the initial pH of the clay (pH = 7–8), which confirms the findings in [3,29]. In mixes with WCP, the pH is significantly increased (pH > 9) due to the hydration processes of the cement. Thus, gypsum solubility increases as well as the surface solubility of fine-particle silicates and aluminates present in the clay, creating favorable conditions for chemical interactions between sois components such as pozzolanic reactions and ettringite formation [12,29].

**Table 3.** pH value of ingredients and mixes.

| Ingredient/Mix | pH, $H^+$ |
|:---:|:---:|
| Quaternary clay (Cl100) | $7.77 \pm 0.21$ |
| Gypsum | $7.32 \pm 0.24$ |
| WCP | $11.78 \pm 0.20$ |
| Water | $6.02 \pm 0.12$ |
| Cl95-G5 | $7.13 \pm 0.16$ |
| Cl90-G10 | $7.10 \pm 0.22$ |
| Cl80-G20 | $7.17 \pm 0.16$ |
| Cl85-G10-WCP5 | $9.16 \pm 0.21$ |
| Cl90-G5-WCP5 | $9.33 \pm 0.15$ |
| Cl95-WCP5 | $9.74 \pm 0.13$ |

Table 4 presents the semi-quantitative analysis of modified soil mixes and their components which are detected by XRD analysis. Apart from the hydration of bassanite, there is no evidence for other chemical reactions and interactions between the clay and the recycled gypsum. As expected, ettringite was formed in the mixes of WCP and recycled gypsum—in modified clay [29]. In those mixes, the quantity of calcium silicate hydrates such as tobermorite increased (from theoretically 0.5% as the share of WCP with 10% tobermorite is only 5% in the different WCP-containing mixes to 0.8%, 2.8% and 3.4%), which confirms the

hydration capacity of non-hydrated cement present in WCP and, probably, the potential of pozzolanic reactions [9,12].

**Table 4.** Percentage (%) of the phases detected by XRD. Semi-quantitative analysis of modified soil mixes and their components.

| Components/Mixes | Main Identified Phases, %wt | | | | | | | | | |
|---|---|---|---|---|---|---|---|---|---|---|
| | Q | F | A | M-IM | D | B | G | T | C-CH | E |
| Recycled Gypsum | - | - | - | - | - | 3.8 | 92.3 | - | 3.9 | - |
| WCP | 28.3 | 33 | 0.0 | 9.9 | 15.0 | - | - | 10.1 | 4.2 | - |
| Cl100 | 43.9 | 41 | 7.4 | 7.8 | - | - | - | - | - | - |
| Cl95-G5 | 44.1 | 34 | 11.2 | 5.2 | - | - | 5.6 | - | - | - |
| Cl90-G10 | 30.2 | 47 | 4.6 | 5.2 | - | - | 12.5 | - | - | - |
| Cl80-G20 | 24.9 | 34 | 8.9 | 7.5 | - | - | 24.7 | - | - | - |
| Cl85-G10-WCP5 | 38.0 | 24 | 5.7 | 14.5 | - | - | 13.1 | 3.4 | - | 1.3 |
| Cl90-G5-WCP5 | 46.6 | 27 | 9.0 | 8.2 | - | - | 5.0 | 2.8 | - | 1.3 |
| Cl95-WCP5 | 50.7 | 33 | 4.1 | 8.1 | 2.1 | - | - | 0.8 | 1.0 | - |

(Q—quartz, F—feldspars, A—amphiboles, M-IM—mica, Illite, montmorillonite, D—dolomite, B—bassanite (semi-hydrate), G—gypsum (di-hydrate), T—tobermorite, C-CH—calcite and portlandite, and E—ettringite).

### 3.2. Physical Properties

Tables 5 and 6 summarize the evaluated physical and engineering properties of the studied soil mixes.

**Table 5.** Physical and engineering properties of clay and recycled gypsum-modified clay mixes.

| No | Characteristic | Method | Cl100 | Cl95-G5 | Cl90-G10 | Cl80-G20 |
|---|---|---|---|---|---|---|
| 1 | Specific density, $\rho_{S,max}$, Mg/m$^3$ | BDS EN ISO 17892-3:2016 [20] | 2.65 | 2.63 | 2.61 | 2.57 |
| 2 | Maximum dry density, $\rho_{d,max}$, Mg/m$^3$ | BDS EN 13286-2:2011 [30] | 1.68 | 1.64 | 1.62 | 1.53 |
| 3 | OMC, $W_{opt}$, % | BDS EN 13286-2:2011 [30] | 17.1 | 17.8 | 19.4 | 22.0 |
| 4 | Porosity, n, % | Calculation based on parameters determined as per BDS EN ISO 17892-1, 2, 3 [19–21] | 38.8 | 39.1 | 41.1 | 39.7 |
| 5 | Internal friction angle, $\phi$, ° | BDS EN ISO 17892-10:2019 [31] | 29 | 27 | 27 | 31 |
| 6 | Cohesion, c, kPa | BDS EN ISO 17892-10:2019 [31] | 34 | 61 | 79 | 60 |
| 7 | Oedometric modulus, $E_{oed100}$, kPa | BDS EN ISO 17892-5:2017 [32] | 3700 | 4000 | 4100 | 4800 |
| 8 | Oedometric modulus, $E_{oed200}$, kPa | BDS EN ISO 17892-5:2017 [32] | 10,843 | 9090 | 10,748 | 15,171 |
| 9 | Oedometric modulus, $E_{oed300}$, kPa | BDS EN ISO 17892-5:2017 [32] | 17,461 | 13,796 | 17,030 | 23,649 |
| 10 | UCS, kPa | BDS EN 13286-41:2021 [33] | 224 | 236 | 210 | 177 |

**Table 6.** Physical and engineering properties of clay and WCP-modified clay mixes.

| № | Characteristic | Method | Cl100 | Cl85-G10-WCP5 | Cl90-G5-WCP5 | Cl95-WCP5 |
|---|---|---|---|---|---|---|
| 1 | Specific density, $r_{S,max}$, Mg/m$^3$ | BDS EN ISO 17892-3:2016 [20] | 2.65 | 2.61 | 2.63 | 2.65 |
| 2 | Maximum dry density, $r_{d,max}$, Mg/m$^3$ | BDS EN 13286-2:2011 [30] | 1.68 | 1.62 | 1.66 | 1.69 |
| 3 | OMC $W_{opt}$, % | BDS EN 13286-2:2011 [30] | 17.1 | 20.5 | 19.5 | 18.3 |
| 4 | Porosity, n, % | Calculation based on parameters determined as per BDS EN ISO 17892-1, 2, 3 [19–21] | 38.8 | 40.5 | 31.9 | 26.1 |
| 5 | Internal friction angle, $\phi$, ° | BDS EN ISO 17892-10:2019 [31] | 29 | 27 | 24 | 28 |
| 6 | Cohesion, c, kPa | BDS EN ISO 17892-10:2019 [31] | 34 | 78 | 96 | 82 |
| 7 | Oedometric modulus, $E_{oed100}$, kPa | BDS EN ISO 17892-5:2017 [32] | 3716 | 3920 | 5220 | 5679 |
| 8 | Oedometric modulus, $E_{oed200}$, kPa | BDS EN ISO 17892-5:2017 [32] | 10,843 | 11,144 | 13,796 | 11,128 |
| 9 | Oedometric modulus, $E_{oed300}$, kPa | BDS EN ISO 17892-5:2017 [32] | 17,461 | 17,082 | 20,091 | 14,863 |
| 10 | UCS, kPa | BDS EN 13286-41:2021 [33] | 224 | 305 | 309 | 331 |

By adding gypsum to the clay, the OMC increases from 17.1% (Cl100) to 22%(Cl80-G20), which indicates that gypsum-modified soil can be compacted to the same degree of compaction at a higher water content (Figure 5), as was reported in [16]. Since the recycled gypsum is predominantly di-hydrate, the "drying-out" effect can be mainly attributed to its water absorption capacity of 2.6 g/L [16]. The addition of gypsum leads to a noticeable decrease in the maximum dry density $r_{d,max}$, more pronounced with high gypsum content (20%), as mentioned in [16]. This result can not only be explained by the lower specific density of gypsum particles, but also by the increase in the porosity (Table 5). So, the recycled gypsum does not considerably improve soil compatibility. The positive effect on compaction of 5% WCP is more definitive in regard to $r_{d,max}$—it remains the same as that of compacted clay, while the porosity is significantly reduced, from 0.39 to 0.26. When a combination of 5% recycled gypsum and 5% WCP is applied, the OMC increases, $r_{d,max}$ is close to that of Cl100 and porosity has an intermediate value of 0.32 (Table 6). The increase in gypsum to 10% worsens the results, except for OMC.

### 3.3. Engineering Properties

Shear parameters in general are of great importance in terms of geotechnical design and consideration, because the shear resistance is highly representative of the bearing capacity of soils.

Figure 6 illustrates the main shear resistance parameters—friction angle ($\phi$) and cohesion I—which were obtained based on the direct shear test. According to [34], the error does not exceed 10% if the horizontal displacement is less than 7.7% of the specimen size, which is respected in our tests (the maximum displacement is 7%). It was established that the friction angle is impacted less impacted by the modifications—it is in the range of 27–28° for the majority of mixes. Within the mix of the highest gypsum content (Cl85-G15),

the friction angle reaches 31°, probably due to the reduction in unbound water content as a result of bassanite hydration.

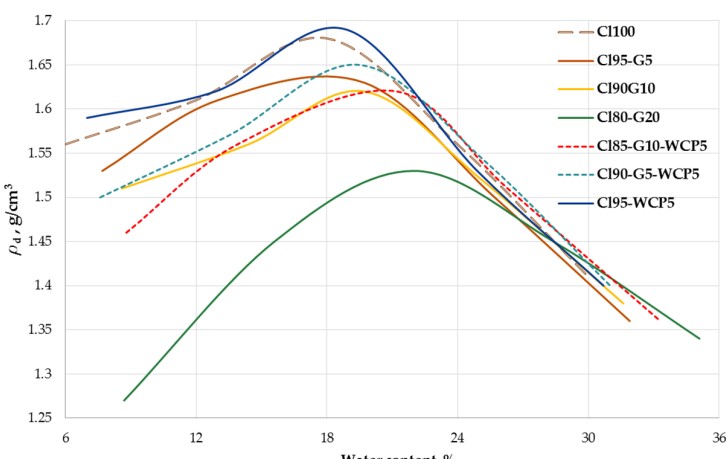

**Figure 5.** Dry density, $\rho_d$, as a function of water content (Proctor curve).

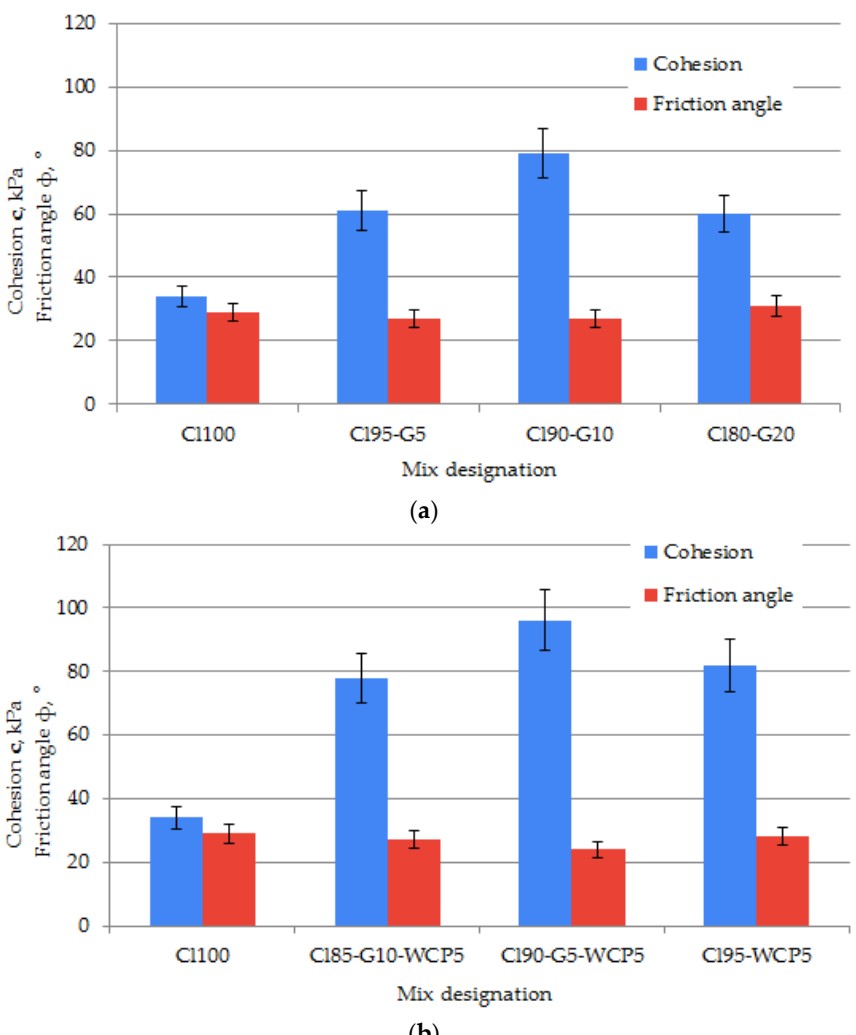

**Figure 6.** Comparison of shear resistance parameters of modified soils: (**a**) modified by recycled gypsum only; (**b**) modified by WCP with or without recycled gypsum.

The modification of quaternary clay by recycled gypsum contributes considerably to soil cohesion—Figure 6a. When compared to that of pure clay mix (Cl100), an increase in the order of 76% and 79% is recorded for mixes Cl80-G20 and Cl95-G5, respectively. The maximum effect of 132% is achieved for mix Cl90-G10. Since no chemical interactions between the clay and the di-hydrate were identified by XRD analysis, it can be supposed that the main reasons for those results are bassanite hydration and physical modifications in the soil structure—coarser particles were added to the clay. However, too small an amount of gypsum (5% in Cl95-G5) does not have capacity for significant modifications, while too high a content of di-hydrates (20% in Cl80-G20) without significant binding capacity, replacing a cohesive soil such a clay, compensates the positive effect of bassanite hydration and, as a result, the cohesion of Cl80-G20 is lower than that of Cl90-G10. This result correlates with the finding of other studies, stating that the maximum cohesion is achieved with gypsum addition of 15% [2]. Moreover, as previously discussed, the increase in gypsum to 20% leads to a smaller maximum dry density of the soil.

The effect of WCP on soil cohesion is greater than that of recycled gypsum—the cohesion of mixes with 5% of WCP is from 1.3–1.8-fold higher than that of pure clay mix. Similar grading modifications of the structure are made as with the recycled gypsum, but it seems that the contribution of the hydration process of non-hydrated cement present in WCP and the ettringite formation in the presence of recycled gypsum (in Cl90-G5-WCP5 and Cl85-G10-WCP5) is very significant.

### 3.4. Oedometer Modulus

The values of the oedometer modulus ($E_{oed}$) as a function of compression loadings for the different soil mixes are presented in Tables 5 and 6. The oedometer compression stress–vertical deformation relationships are plotted in Figure 7.

Compression behavior is influenced only by the highest gypsum content (20%) in the soil (Cl80-G20), where the lowest deformability of recycled gypsum-modified mixes is observed. A similar result is reported in [3] in regard to the laterite clay modification by gypsum. The deformability of mixes with 5% and 10% of recycled gypsum is practically identical to that of the clay. The less deformable mixes are those with 5% WCP without or with a small amount of gypsum (5%), i.e., Cl95-WCP5 and Cl90-G5-WCP5. For those mixes, the presence of larger crystals of portlandite, calcite and ettringite might explain their good behavior—Table 4. Those mixes are characterized by the lowest porosity—Table 6.

The highest oedometer modulus at the lower stress level ($E_{oed100}$) is determined with the Cl95-WCP5 mix—Table 6. It is more than 50% higher than the $E_{oed100}$ of pure clay. With the increase in stresses (200 kPa), the highest $E_{oed200}$ is more than 15 000 kPa, achieved by the mix with 20% of gypsum (Cl80-G20), Table 5, followed by the mix Cl90-G5-WCP5 (ca. $E_{oed200}$ = 13,800 kPa) containing 5% of WCP and 5% of gypsum—Table 6. Further, with the load increase, the $E_{oed}$ of Cl80-G20 continues to increase (ca. $E_{300}$ = 24,000 kPa), but the superiority in terms of the non-modified soil (ca. $E_{300}$ = 17,000 kPa) is reduced to 35%—Table 5. From the WCP-modified mixes, only Cl90-G5-WCP5 has a higher $E_{oed}$ (ca. $E_{oed300}$ = 20,000 kPa). The mix with Cl85-G10-WCP5 possesses an $E_{oed300}$ in the same range as the clay, while the $E_{oed300}$ of the C95-WCP5 mix is 15% lower than that of the clay—Table 6. It must be mentioned that the $E_{oed}$ of WCP-containing mixes has the potential to increase over time with further development of hydration, pozzolanic reactions, ettringite formation and carbonation processes [3,16,29].

### 3.5. Unconfined Compressive Strength

The average values of the measured UCS and their standard deviations are illustrated in Figure 8. Soil modification by recycled gypsum does not practically affect the UCS of soil. As mentioned in [3], a small amount of gypsum could contribute marginally to the UCS because of the re-arrangement of soil particles, but a further increase in gypsum content does not amplify the rearrangement effect. When the gypsum content reaches 20% (Cl80-G20 mix), the average UCS is reduced by 21% compared to that of non-modified clay.

These results take into consideration that there are no interactions between the recycled gypsum and the clay in a non-alkaline environment (pH of those mixes is around 7) and the very low strength of di-hydrates in moist conditions, due to gypsum solubility.

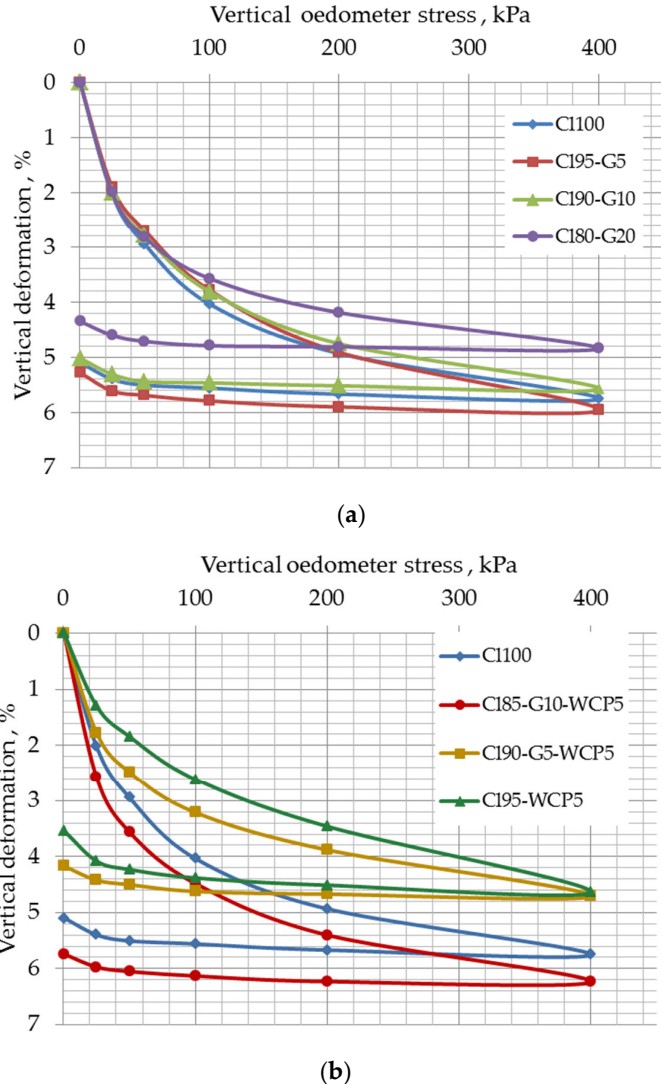

**Figure 7.** Compression stress–vertical deformation relationships for the different mixes obtained by the oedometer test: (**a**) modified by recycled gypsum only; (**b**) modified by WCP with or without recycled gypsum.

The binding effect of WCP is confirmed—the UCS of mixes with 5% of WCP increases by up to 48%, which is in good correlation with other studies. For instance, in [29], a modification by 1.5% of Portland cement and 5% of recycled gypsum semi-hydrates leads to an increase of 95% in the compressive strength of a clayey soil after 3 days of curing.

Although all WCP-containing mixes definitely have a higher UCS than the non-modified soil, the biggest increase is achieved with the mix without recycled gypsum (Cl95-WCP5), which means that the contribution of ettringite is smaller than that of the products of cement hydration. The average UCS values for 5% WCP-modified soils with or without recycled gypsum vary from 300 to 330 kPa (Table 6) and thus they are similar to UCS (320 kPa), obtained by using 3% of Portland cement and 5% gypsum semi-hydrates for a 3-day curing time [29].

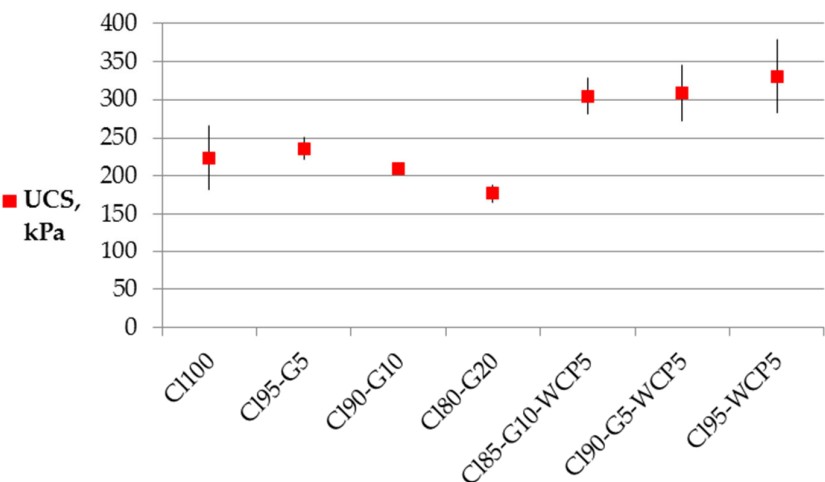

**Figure 8.** Average values of UCS and their standard deviations for different soil mixes.

An increase in the UCS of WCP-containing mixes can be expected over time with further development of hydration and pozzolanic reactions. Pozzolanic reactions might require some time (several months) to develop [9,12,17], while the UCS was only determined 3 days after sample preparation and curing in sealed conditions.

## 4. Conclusions

In situ soil stabilization is the best option to solve the workability, bearing capacity and deformability issues of clayish foundations. The mechanical recycling (grinding into powder) of plasterboard and concrete CDW offers a great possibility to replace natural gypsum and binders such as cement and lime in the soil stabilization process.

The stabilizing effect of recycled gypsum di-hydrates on quaternary clays of the region of Sofia is relatively limited to improvement in soil compaction. A gypsum content of 10% seems to be the optimum level in terms of improvement in other engineering characteristics such as soil cohesion and the oedometer modulus in the non-saturated state of the soil. There is no evidence of chemical interactions between mechanically recycled gypsum and quaternary clay. The positive effect of gypsum stabilization is rather due to the water absorption by gypsum powder and particles rearrangement of stabilized clay mixes.

Waste concrete powder (WCP), obtained by mechanical recycling of waste from reinforced concrete sleepers, seems to be an active compound for clayey soil modification and has a positive effect similar to that of Portland cement—by adding 5% only of WCP, a significant reduction in deformability and improvement in soil cohesion and the UCS of quaternary clays can be achieved. When 5% recycled gypsum is added to the system "clay-WCP", because of ettringite formation and the physical effect of gypsum powder, some engineering properties such as cohesion are further improved. Therefore, the combination of WCP and recycled gypsum can be appropriate in some particular cases of soil stabilization.

In conclusion, recycling of concrete and plasterboard CDW into powder is a technically feasible solution for improvement in soil foundation at a reduced environmental footprint, because of the decreased $CO_2$ emissions, use of local materials and deviation of CDW from landfilling or down-cycling.

**Author Contributions:** Conceptualization, R.Z. and D.E.; methodology, R.Z. and N.K.; software, D.E.; validation, R.Z., D.E. and N.K.; formal analysis, D.E., N.K. and T.S.; investigation, R.Z. and D.E.; resources, R.Z., D.E., N.K. and T.S.; writing—original draft preparation, D.E. and N.K.; writing—review and editing, R.Z.; visualization, D.E. and N.K.; project administration, R.Z.; funding acquisition, R.Z. All authors have read and agreed to the published version of the manuscript.

**Funding:** The research presented in this paper was carried out in the framework of the Clean & Circle Project BG05M2OP001-1.002-0019: "Clean technologies for sustainable environment—waters, waste,

energy for circular economy", funded by the Operational Program "Science and education for smart growth" 2014–2020, co-financed by the European Union through European structural and investment funds. The authors acknowledge the financial support for the project.

**Institutional Review Board Statement:** Not applicable.

**Informed Consent Statement:** Not applicable.

**Data Availability Statement:** Not applicable.

**Conflicts of Interest:** The authors declare no conflict of interest.

## Abbreviations

| | |
|---|---|
| A | Amphiboles |
| B | Bassanite |
| BDS | Bulgarian State Standard |
| c | Cohesion |
| C-A-H | Calcium aluminate hydrate |
| C-A-S-H | Calcium aluminate silicate hydrate |
| C-CH | Calcite and portlandite |
| CDW | Construction and demolition waste |
| CH | Calcium hydroxide |
| Cl | Clay |
| C-S-H | Calcium silicates hydrate |
| D | Dolomite |
| DST | Direct shear tests |
| E | Ettringite |
| EN | European Standards |
| $E_{oed}$ | Oedometric modulus |
| $\phi$ | Friction angle |
| F | Feldspars |
| G | Gypsum |
| GHG | Greenhouse gases |
| Ip | Plasticity index |
| ISO | International Organization for Standardization |
| M-IM | Mica, illite, montmorillonite |
| n | Porosity |
| OMC | Optimum moisture content |
| pH | Potential of hydrogen |
| Q | Quartz |
| **r** | Bulk density |
| **r**$_d$ | Dry density |
| **r**$_{d,max}$ | Maximum dry density |
| **r**$_S$ | Specific (true) density |
| SiCl | Silty clay |
| T | Tobermorite |
| UCS | Unconfined compression strength |
| WCP | Waste concrete powder |
| $W_L$ | Liquid limit |
| Wp | Plastic limit |
| XRD | X-ray diffraction phase analysis |
| $\phi$ | Internal friction angle |

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
