# Peer review of "Modification of Quaternary Clays Using Recycled Fines from Construction and Demolition Waste"

_processes, doi:10.3390/pr10061062_

Round 1

Reviewer 1 Report

This paper presented an interesting study on developing sustainable recycled finds for building repairs, reinforcement and possible foundation retrofit. It is an important topic in supporting circular economics in construction. There are a few minor comments for the authors to consider:

  1. Page 3, 108-112, this is the only part the authors talked about the aim(s) of this study. It would be better to summarise a list of slightly more details of objectives what they are going to achieve plus a justification of the methods below when compared to the previous studies above.

  1. Figure 1,3,4 the text is too small. Better to enlarge them. Ideally, size, font etc can be unified in all Figures.

  1. The conclusion is quite a detailed one, which is ok but it would be better to summarise a list of key findings of this study.

  1. Some written English can be improved, such as “Very attractive opportunity is the utilization of industrial waste and by-products 50 (such as bottom ashes, fly ashes, slag and phosphogypsum), but … ”. It would be better to get a native speaker to proofread it.

Overall speaking, it is an outstanding work, after minor correction, I recommend it to be published on the journal

Reviewer 2 Report

This work (Manuscript ID: processes-1743260) titled “Modification of quaternary clays by using recycled fines from construction and demolition waste (CDW) assessed the possibility to use recycled fines from construction and demolition waste, instead of ordinary binders or industrial waste, in the stabilization of quaternary clay, which is of great significance for improving the mechanical properties of quaternary clays, reducing the impact of soil stabilization on the environment, and improving the CDW recovery of concrete and gypsum. This manuscript is well organized. I suggested this manuscript can be accepted after minor revision. The following questions should be addresses.

  1. Some grammar errors need to be revised: eg., “Particle size distribution of Quaternary clays is determinate by EN ISO 17892-4:2016 129 – [22] and is illustrated on Figure 1” Here, the “determinate” should be corrected as “determinated”.
  2. In Section 2.5.2, the number of pH measurements is not accounted for in the manuscript, and the deviation of the data is missing (see Table 3). The authors should account for the number of measurements and add the standard deviation of the data.
  3. For Figure 7, the author should describe the test number and added the error bar.
  4. The conclusions are too long and needs to be simplified. The conclusion should describe the main findings of the study, its advantages over previous studies, and the outlook for the study.
